# Fire Characteristics of Upholstery Materials in Seats

**DOI:** 10.3390/ijerph17093341

**Published:** 2020-05-11

**Authors:** Linda Makovická Osvaldová, Iveta Marková, Miroslava Vandlíčková, Stanislava Gašpercová, Michal Titko

**Affiliations:** 1Department of Fire Engineering, Faculty of Security Engineernig, University of Žilina, Univerzitná 1, 010 26 Žilina, Slovakia; linda.makovicka@fbi.uniza.sk (L.M.O.); miroslava.vandlickova@fbi.uniza.sk (M.V.); stanislava.gaspercova@fbi.uniza.sk (S.G.); 2Department of Crisis Management, Faculty of Security Engineernig, University of Žilina, Univerzitná 1, 010 26 Žilina, Slovakia; michal.titko@fbi.uniza.sk

**Keywords:** initiation sources of ignition, cigarettes, matches, upholstery materials in seats

## Abstract

The article deals with selected upholstery flammability test materials that, in the case of fire, can cause fire spread. For the research, frequently used materials for upholstery based on polyester were utilized: imitation leather, suede, and microplush. Initiation of initiating spontaneous flammability with various sources of ignition were measured including a smoldering cigarette and a match flame. Results were measured as smoldering time and length of the burnt-though sample. Upholstery materials are an integral part of seat construction. To be used in transport, upholstered material must meet safety measures such as the strength, sanitariness, and fire resistance. All tests were performed in accordance with applicable technical standards. Impact assessment of the sample (weight) on “smoldering time” and “length of degradation” was carried out using an ANOVA. Significant differences in length of degradation was observed between samples. Tests cannot provide reliable information about the flammability course of the final product. Upholstery is composed of external covering, of inner liner, and padding. Results of the research presented in this paper indicate the need to continue the research in a broader aspect.

## 1. Introduction

Materials and their development depend on their purpose, in order to find the ones with excellent properties that best serve the intended purpose. The development method is based on tests. In aviation, the use of composite materials is growing thanks to their practical advantages [1] such as: building larger window openings; achieving a lower resistance of air flow, which contributes to a reduction in fuel consumption; and reduction in the total number of parts and components. On one hand, there are efforts to reduce the amount of material, used but on the other hand, it is vital to consider an increased risk of fire initiation of the given materials.

The organization laying down the general rules for the fire safety of the materials used in civil aviation is the FAA (Federal Aviation Administration). FAA regulations technically apply to the aircraft designed and manufactured in the USA, but they are applied in general in the whole aviation sector [2]. FAA has drawn up fire test procedures, in regulation FAR 25.853 Flammability Requirements for Aircraft Seat Cushions [3], to test flammability and fire resistance. FAR 25.853 [3] sets out the major fire properties of materials, namely, the total heat release rate, heat release rate, smoke release, and temperature of surfaces [2].

According to the regulations in civil aviation, fire, explosion, smoke, and toxic or harmful fumes, even after putting out a fire, are considered states of emergencies [2,4,5,6].

The FAA has also drawn up fire test procedures, outlined in regulation FAR 25.853 [3], to test materials’ flammability and fire resistance. FAR 25.853 sets out the major fire properties of materials (Table 1), such as the total heat release, heat release rate, and smoke release [2].

The limit values presented in Table 1 are specified to ensure that the material used in the aircraft does not contribute to the development and spread of fire during the first five minutes after landing [7].

The Federal Aviation Administration (FAA) aimed to create a database of the materials tested that have been used or are in the development stage for their use in civil aviation. The database is to provide air safety authorities, engineers in the aviation industry, and aviation technology operators with information on flammability and fire characteristics of the tested materials [2].

Fire-fighting properties in the FAA database determined using the cone calorimeter test are only provided for the heat flow of 50 kW/m^2^, because most available data on fires of composite materials are designed for fire testing for the given heat flow (Table 1). This heat flow creates the surface temperature of maximum 700 °C on the polymer composite, which is equivalent to a fire of medium intensity in the aircraft [2].

The FAA database also offers additional fire properties of (composite) materials such as: time until ignition, limit oxygen index, maximum heat release rate, average heat release rate, total heat release, flame spread rate, and production of smoke and combustion products.

Upholstered seats represent a part of the interior equipment of an aircraft. There are separate tests relating to these materials [8,9].

Wang et al. [10] simulated the conditions of fire in the cargo area of an aircraft. They studied behavior characteristics of the selected materials in a partially closed chamber and in an alpine laboratory for the same low atmospheric pressure. The authors observed the control effects of exhaust air and oxygen concentration. Experiments with fire using n-heptane were carried out separately in a low-pressure chamber Langfang (altitude 50 m) and in airport laboratories Kangding (altitude 4290 m) at the air pressure of 60 kPa according to ISO-9705 [11]. The main characteristic parameters were burning rate, flame temperature, and radiation heat flow. Increased oxygen concentration caused the burning rate of the material to increase and burning time to decrease. A higher amount of air absorbed, or higher oxygen concentration, could increase the fire intensity in the cargo area and worsen the flame spread rate.

### 1.1. Textile Materials in Seats

Currently, upholstery forms an integral part of seat construction. In the industrial sector, upholstered material must meet safety measures such as the strength, sanitariness, and fire resistance. Upholstery fabrics, which belong to the category of combustible substances, are a part of the upholstered product. Therefore, it is necessary to implement measures to increase fire safety of these textiles. Seats are a part of an aircraft and are covered with upholstery material. Upholstery is the surface treatment of materials and walls using textile, leather, leatherette, netting, or other natural and synthetic materials [8]. Upholstery is composed of external covering (outer cover), of inner liner (inner cover; coating thicker than 2 mm), and padding (filling).

Cushioning belongs to the category of materials that are easily flammable, such as latex foam, polyurethane foam, etc. These materials react dangerously in contact with fire, such that they melt, run, and release toxic residues.

Methods determining the ability of materials to ignite is based on the setting of boundary conditions in which we observe flare-up, i.e., ignition.

The basic parameters to be assessed when observing the behavior of the selected upholstery materials are: [8] time of the flame exposure (flame application time) which is the time a test sample is exposed to ignition flame; and time of flame spread (flame spread time), which is the time the flame covers a certain distance onto the burning material under the specified test conditions.

### 1.2. Sources of Combustion Initistion

All aspects of a fire have to test in the field of fire protection. One of the basic conditions of combustion is the presence of an effective initiation source. To initiate burning, the initiation source must have a sufficient amount of energy. After initiating the burning process, to continue burning it needs sufficient heat output either from the combustion zone itself or in cooperation with other heat sources. A necessary condition of initiation, as well as of continuous burning, is the existence of combustible file, i.e., a suitable quantity ratio of flammable substances and oxygen [7,12,13]. Based on the initiation source, we can identify three types of fire initiation processes [14]: (1) spontaneous ignition caused by an external radiant heat source; (2) initiation caused by external source of ignition (open flame, spark); and (3) spontaneous ignition without any external heat source (self-ignition, chemical reaction). Conditions for burning initiation are limited in particular by the concentration of fuel and oxygen and the source of ignition (flame, radiant heat, spark, etc.) [15,16,17,18]. Initiation source always refers to a particular combustive or an explosive system [19,20,21,22].

Initiation energy is fed to the combustive material by an external initiation source. It is a so-called “external” initiation. However, there are substances that are able to self-activate as a result of their particular instability (either the chemical composition or physical properties).

Such substances are referred to as self-ignitive [23]. Initiation sources are divided as follows [24]: hot surfaces; flames and hot gases (including hot particles); mechanical sparks, electrical devices; stray voltage and cathodic protection against corrosion; static electricity; lightning; high-frequency (HF) electromagnetic waves ranging from 104 Hz to 31,012 Hz; electromagnetic waves from 31,011 Hz to 31,015 Hz; ionizing radiation; ultrasound; adiabatic compression, shock waves, and exothermic reactions; and including auto-ignition of dust.

The specific sources of ignition for the manufacture of upholstered sitting furniture are smoldering cigarette and equivalent of flame by matches. The energy released from the cigarettes proves itself as not very effective, with no flame initiation in most cases [25]. The cigarette burning is, in the process of smoldering, accompanied by the release of harmful products. Their characteristics are presented in the works of [26].

Stoliarov [27] explains the processes as follows: “The pyrolysis region, dominated by anaerobic decomposition, provided gaseous fuel, the ignition of which resulted in the transition. The smoldering region, dominated by oxidation reactions at the solid–gas interface, generate the heat necessary to maintain the pyrolysis process and ignite the gaseous fuel” [27].

We need to note that objections to the existing methods of the tests of upholstery material have been made. In 2012, Lloyd treated this issue [28] while presenting the diversity of cigarette position and the time of application on wheelchair materials.

The aim of this article was to analyze and compare of the effect of initiation sources (cigarettes and matches) on the upholstery materials, namely surface materials of the upholstery set and impact assessment of the materials on the degradation processes.

## 2. Methodology

### 2.1. Samples

Upholstered furniture tests were carried out using three basic samples (Figure 1) which are often used and are easily available. The samples have been cut into desired dimensions in accordance with the European norms in force. Each test specimen has a different composition and surface (Table 2).

### 2.2. Flammability Assessment of Upholstered Furniture Using a Smoldering Cigarette

European standards specify the test method for flammability tests of material combinations, such as upholstery covers and paddings/stuffings, used for the manufacture of upholstered sitting furniture, where a smoldering cigarette represents the source of ignition. The test determines only the flammability of material combinations, which the upholstered sitting furniture consists of, and does not determine the flammability of individual materials used for its manufacture. The test indicates its flammability but cannot provide reliable information about the flammability course of the final product (furniture) [8].

In the current experiment, a set of upholstery materials was exposed to the ignition source, a smoldering cigarette. The materials were arranged so that the set matches with the place between the bottom of an armchair and the armrest. Flammability of the set was determined using a cigarette. The method tests the flammability of the whole system i.e., upholstery cover, inter-liner, paddings, and stuffings etc. in the arrangement suitable for the test equipment [8].

Within 20 minutes after taking the materials out of the conditioning atmosphere, a cigarette was lit and the air was absorbed through it until the end of the cigarette was visibly burning. During this stage, at least 5 mm but no more than 8 mm of the cigarette length was spent [6]. The burning cigarette was positioned along the contact line between the horizontal and vertical part of the test assembly, so that the cigarette was at least 50 mm from one of the edges or from any area potentially damaged by the previous test. At the same time, the timer was launched [8].

The burning course was monitored and any traces of sustained smoldering or flame combustion of the padding or cover was recorded. If sustained smoldering or flame combustion of the layers of the upholstery occurred, the test assembly was put out and the information was recorded; the time between placing the cigarette onto the test assembly and extinguishing was also recorded. If ignition by sustained smoldering or by flame combustion did not occur, or the cigarette did not burn its full length, then the information was recorded and the test was repeated once again using a new cigarette at 50 mm distance from the area potentially damaged by the previous test [8].

If no ignition caused by the sustained smoldering or flame combustion occurred, or if the cigarette did not burn its full length during the second test, then the data was recorded and a final inspection was made [8]. The cleaned test equipment was unfolded and both the upholstery cover and the inter-liner (if there is any) were fixed to a bar. Stuffings and paddings were placed below the inter-liner and were fixed into frames. The frame covered by the upholstery cover and 20 mm of the cover was sticking out. By using at least four clips for each side, the cover was fixed onto the frame on the top and bottom part of the frame. Clips were at least 60 mm long (Figure 2).

### 2.3. Flammability Assessment of Upholstered Furniture Using a Match Flame

In this test, upholstery materials (Figure 2) were exposed to the flame of a gas burner, which simulates that of a match. The upholstery materials were arranged so that it stylistically matched the connection between a seat and backrest, as it is the case in ordinary sitting furniture. The test procedure was the same as for cigarette EN 1021-2:2014 [9]. The difference lies in the time of flame exposure (10 ± 1 s) and the application of the gas burner as the equivalent of a match [9].

During the experiments, we followed the standard criteria specified, which are evaluated in Table 3 for the smoldering cigarette initiator and in Table 4 for the igniting equivalent. The results of the experiments include visual observation of thermal changes over time and experimental data of smoldering time (min) and length of the burnt-though sample (cm).

## 3. Results and Discussion

### 3.1. The Results for a Smoldering Cigarette

The Smoldering cigarette had an impact on all of the tested samples (Table 3). During the experiment, there was thermal degradation of the surface of the test materials (Figure 3), however, the whole test assembly did not burn completely, and it did not burn for longer than 60 seconds (Table 3). The experiment realized by an initiation source (cigarette) was carried out in the form of flameless burning, i.e., smoldering, except for Sample 3.

Sample 1 and Sample 2 were smoldering flamelessly at the point of contact between the cigarette and the material (Figure 3a,b). In Sample 3 experiments (Figure 3c), gradual smoldering changed into flame burning of the upholstery, which was dangerously extending to the edges, and consequently, the burning parts started to drip off. In all the cases, the cigarette smoldering occurred at different time intervals, which is reflected in the size of the area burned (Table 3).

During the test, Sample 1 (Figure 3a) only burnt through its thickness, such that the whole assembly did not burn down according to the criteria in Table 2. Based on the behavior of Sample 1 and its chemical composition, we can see that polyvinyl chloride (PVC) material does not withstand heat, rather it melts and subsequently drips off at low temperatures [32,33,34]. On the other hand, PVC is a poor heat conductor, i.e., the given process takes place only in contact with the initiator [35,36,37]. Sample 2 (Figure 3b) smoldered only at the point of contact between the cigarette and the material. Thermal degradation did not spread, and the material did not burn through (Figure 3bIII). The smoldering cigarette on the Sample 2 (Figure 3bI,II) was not the cause of the fire; neither smoldering not fire spread occurred (Figure 3a). Results of thermal degradation Samples 2 showed influence of additives, specifically 20% acrylic. Results of thermal degradation of Samples 2 showed influence of additives, specifically 20% acrylic. The experiments were realized in the atmospheric same conditions and the time is the important output parameter. Figure 3 offers photos with selected experiments. Each picture was made at actual time.

Sample 3 contains cotton in its structure (Table 2). Putted cigarette (Figure 3cII) started initiation quicker than Sample 1 (Figure 3aI,II) and Sample 2 (Figure 3bI,II). Wanna [38] points out the flammability of cotton and emphasizes the need for its retarding treatment, e.g., using appropriate salts. The given quantity of cotton influenced the behavior of the material in the experiment (Figure 3c).

### 3.2. The Ignition Using an Equivalent to a Match

The experiments were more dynamic for a match than in the case of a cigarette. The criteria specified in [8] were not fulfilled (Table 4). In each experiment, the sample ignited flamelessly. The time of flame exposure was 10 ± 1 s. During the experiment, grayish-black smoke was created. The intensity of burning grew as the experiment progressed. When the flame was removed from upholstery seats, the samples stopped their self-burning (Figure 4).

Sample 1, after having been exposed to the flame of the gas burner for 10 s, ignited all over the surface. The experiment using Sample 2 lasted 156 s; measured length of the burned area was 12.7 cm (Figure 4b). The burning did not expand dangerously, there was no smoldering (Figure 4). Sample 3 burned down.

### 3.3. Mathematical and Statistical Processing and Evaluation of Results

The results of the experiments (Table 5, *N* = 5) were subjected to a statistical analysis, which consisted of two parts: (1) comparing the differences in the samples (analysis of differences) and then selecting the most appropriate sample for the purposes pursued; (2) investigating the impact of the “weight” on the monitored variables (analysis of dependence).

First, we investigated whether there were significant differences between Samples 1, 2, and 3 with respect to each selected variable individually (experimental parameters from Table 3 and Table 5). A *t*-test is generally used to compare the means of two groups. Therefore, *t*-tests were performed separately for the parameter “time of smoldering” and separately for “length of degradation” for all sample combinations. For the given conditions, the results suggest that there are statistically significant differences between samples (*p* < 0.01).

As the samples show differences with respect to individual parameters, the main aim of the analysis was to find out whether there are significant differences between samples considering both parameters at the same time. Therefore, we test the alternative hypothesis H1 against the null hypothesis H0. H1 was that samples divided into three groups differ significantly in terms of “time of smoldering” and “length of degradation”. Conversely, H0 was that there is no statistically significant difference between samples with respect to “time of smoldering” and “length of degradation”.

The differences between samples of the selected textile upholstery materials were statistically evaluated using double factor analysis of variance (ANOVA) with replications. The results are shown in Table 6.

Based on the results of the ANOVA analysis, we can see that the F values were higher than the F critical values and *p*-values were lower than alfa 0.01. Therefore we have statistically significant results. Based on these results we reject the hypothesis H0, and therefore claim that the samples differ significantly with respect to the parameters examined.

Based on the results obtained from the analysis of differences and experiment evaluations, we can conclude that Sample 2 has the most appropriate behavior and attributes, as the given sample had the shortest time of smoldering (Table 5).

For an evaluation of the impact of the sample (weight in particular) on “time of smoldering” and “length of degradation”, the results were subjected to the dependence analysis. Pearson’s and Spearman’s correlation coefficients were used to explore the relations between: (1) the dependent variable “time of smoldering” and independent variable “weight”; and (2) the dependent variable “length of degradation” and independent variable “weight”.

Regardless of the sample type, the weight of the samples had a statistically significant effect on “time of smoldering” (*p* < 0.01), and at the same time, this dependence was strong, as confirmed by Pearson’s coefficient r = 0.917 (Table 7). No dependence was confirmed between weight and length of degradation (*p* > 0.05; r = 0.163; Table 7).

It can be concluded that the “weight” only affects the “time of smoldering” variable. As “weight” increases, do does the “time of smoldering”. Although Samples 2 and 3 had the same weights, the previous analysis of variance showed that they differed significantly in parameter “time of smoldering”. The time of smoldering was lower for Sample 2, indicating its more suitable properties.

## 4. Conclusions

Based on the experiments carried out, we have formulated the following conclusions:
Experiment with a smoldering cigarette: Sample 1: the test sample burned through, without the climate change presence of a flame. Sample 2: No active smoldering occurred. It burned through its thickness and the cigarette went out. Sample 3: burned through its entire thickness.Experiment using an equivalent to a match flame: None of the selected upholstery material complies with the EN 1021-2:2014 conditions. Sample 1: the whole test assembly burned through (including the stuffing: PUR foam) and flame burning spread across the whole surface of the test sample.Cushioning material is ignitable already in contact with a minor source of ignition and has a tendency to burn with an increasing intensity.The most appropriate parameters were observed for Sample 2 (PES/ACRYLIC).

The individual files of experiments of flammable upholstered seats are realized by Slovak research institutes (for example rate of heat release, amount and type of compounds released, flame propagation rate). Our study, which tested cigarette and simulated equivalent of flame indicates flammability but cannot provide reliable information about the flammability course of the final product (furniture) which have to be added as other fire parameters. The research presented in this article needs to be continued in the future.

## Figures and Tables

**Figure 1 ijerph-17-03341-f001:**
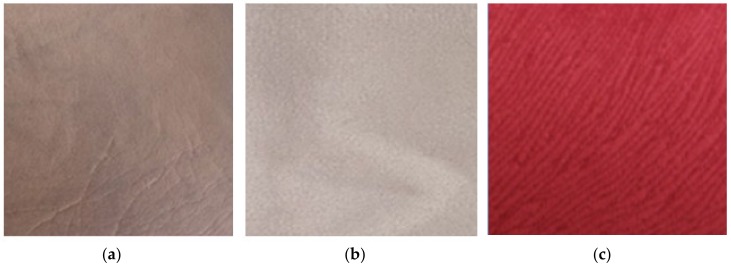
Pictures of the three samples. (**a**) Imitation leather, (**b**) suede, and (**c**) microplush.

**Figure 2 ijerph-17-03341-f002:**
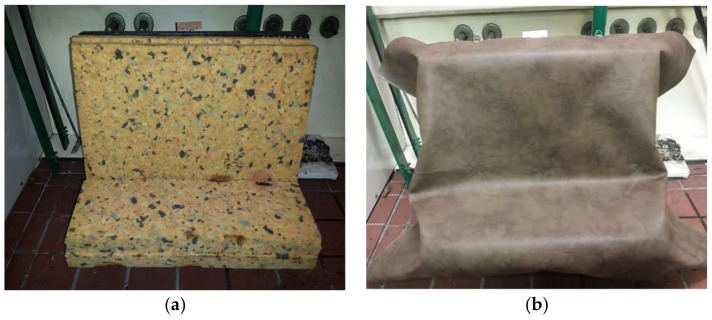
Test equipment. (**a**) underlying panel and (**b**) upholstery for the experiment.

**Figure 3 ijerph-17-03341-f003:**
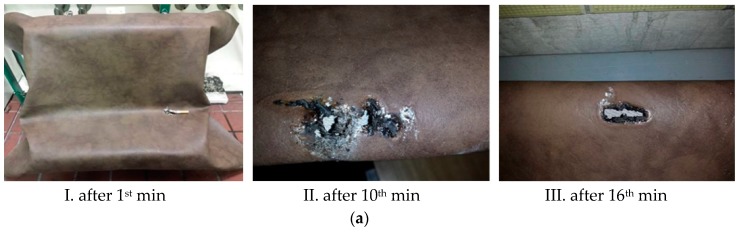
The experiment using a burning cigarette on (**a**) Sample 1, (**b**) Sample 2, and (**c**) Sample 3.

**Figure 4 ijerph-17-03341-f004:**
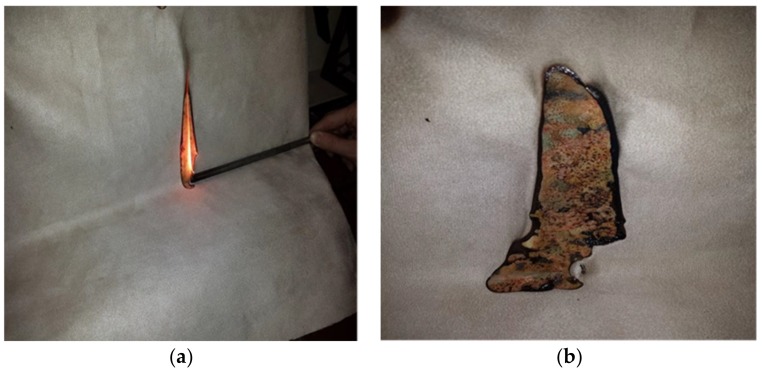
The experiment with Sample 2 using an equivalent to a match. (**a**) Ignition by the equivalent of match, and (**b**) residual after experiment.

**Table 1 ijerph-17-03341-t001:** Fire safety criteria of materials for aviation sector adjusted to the specification FAR 25.853 [3].

Parameter	Criteria
Ignition and spread of flame	Using Bunsen burner test
Heat release for non-metallic materials	Cone calorimeter test for the heat flow of 35 kW.m^2^
Fire intensity (heat flow)	
Total HRR of the test material	Lower than or equal to 65 kW.m^−2^ for two minutes
Maximum HRR (Heat Release Rate)	Lower than or equal to 65 kW.m^−2^ in five-minute test
Specific smoke value (Ds)	Lower than or equal to 200 for four minutes
Fire properties using cone calorimeter	
Heat flow	50 kW/m^2^
Surface temperature on polymer composite	Maximum temperature approximately 700 °C

**Table 2 ijerph-17-03341-t002:** Technical parameters of the upholstered material.

Technical Parameters	Upholstered Material from Material Safety Data Sheets (MSDSs)		
	Sample 1 (Figure 1a) [29]	Sample 2 (Figure 1b) [30]	Sample 3 (Figure 1c) [31]
Color	Grey	Light	Red/Wine
Quality	Imitation leather	Suede	Microplush
Composition—surface	Fabric ^1^: 100% PES	80% PES	98% PES
	100% PVC	20% acrylic	2% cotton
Width (cm)	140	150	140
Weight (g.m^−2^)	390	280	280
Thickness (mm)	(0.70 ± 0.1)	(0.65 ± 0.1)	(0.70 ± 0.1)
Martindale ^2^	60,000 MD	45,000 MD	50,000 MD
Note	No surface treatment, clean with natural texture	Leather with a protective layer ensures greater resistance against sunlight, water and dirt	Covering material—microplush—with a delicate print

^1^ manufactured using indirect application of polyvinyl chloride onto the primer (polyester); ^2^ Martindale method simulates natural wear of a seat cover, in which the textile sample is rubbed against a standard abrasive surface with a specified force; (the most popular method for wear resistance tests) PES, polyester; PVC, polyvinyl chloride.

**Table 3 ijerph-17-03341-t003:** Evaluation tests with a smoldering cigarette according to the given criteria [8].

Criteria		Sample 1	Sample 2	Sample 3
Smoldering	Burning spreading dangerously	No	No	Yes
	Test assembly has burnt out	No	No	No
	Test assembly has burned through by smoldering towards edges	No	No	No/Yes
	Test assembly has burned through by smoldering in its thickness	Yes	Yes	Yes
	Smoldering of the test assembly for more than 60 s	No	No	No
	Presence of active smoldering during the final inspection	No	No	No/Yes
Burning	Occurrence of flames	No	No	No
Experimental parameters				
Smoldering time (min)		15.788 ± 0.443	11.83 ± 0.408	13.212 ± 0.137
Length of the burnt-though sample (cm)		11.856 ± 0.343	6.156 ± 0.328	15.01 ± 0.142

**Table 4 ijerph-17-03341-t004:** Evaluation tests using an equivalent of a match flame according to the given criteria [9].

Criteria	Sample 1	Sample 2	Sample 3
Fire spreading dangerously	Yes	No	Yes
Burned out test assembly	Yes	No	Yes
Burned area up to the edges	Yes	No	Yes
Burned in its entire thickness	Yes	Yes	Yes
Flame burning for more than 120 s	Yes	Yes	Yes

**Table 5 ijerph-17-03341-t005:** Results of Experiments (*N* = 5).

Sample (Content)	Experiment No.	Weight (g)	Time of Smoldering (min)	Length of Thermal Degradation (cm)
1 (PES/PVC)	1	390	15.12	12.23
	2	390	16.42	11.32
	3	390	15.55	12.04
	4	390	16.07	11.59
	5	390	15.78	12.10
2 (PES/ACRYL)	1	280	12.23	5.80
	2	280	11.32	6.50
	3	280	12.05	5.90
	4	280	11.35	5.98
	5	280	12.20	6.60
3 (PES)	1	280	13.25	15.05
	2	280	13.05	15.10
	3	280	13.11	14.80
	4	280	13.45	15.20
	5	280	13.20	14.90

**Table 6 ijerph-17-03341-t006:** Results of two-factor ANOVA with replication.

Source of Variation	SS	df	MS	F	*p*-Value	F Critical
Sample	50.80405	1	50.80405	387.7432	10.1016/5.5 × 10^−15^	4.259677
Columns	165.3223	2	82.66114	630.8807	1.79 × 10^−21^	3.402826
Interaction	76.41521	2	38.2076	291.6054	1.45 × 10^−17^	3.402826
Within	3.1446	24	0.131025			
Total	295.6861	29				

**Table 7 ijerph-17-03341-t007:** Correlations between “weight” and “time of smoldering” or “weight” and “length of degradation”.

Criterion	Correlations	Time of Smoldering	Length of Degradation
weight	Pearson correlation	0.917 **	0.163
	Significance (2-tailed)	0.000	0.561
	*N*	15	15
	Spearman’s rho correlation coefficient	0.818 **	0.000
	Sig. (2-tailed)	0.000	1.000
	*N*	15	15

** Correlation is significant at the 0.01 level (2-tailed).

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
