# Peer review of "Fire Characteristics of Upholstery Materials in Seats"

_ijerph, 2020, doi:10.3390/ijerph17093341_

Round 1

Reviewer 1 Report

1) Text (in the previous version):

„Wang et al. [24] simulated the conditions of fire in the cargo area of an aircraft. He studied behavior characteristics of the selected materials in a partially closed chamber and in an alpine laboratory for the same low atmospheric pressure. He observed the control effects of exhaust air and oxygen concentration. Experiments with fire using n-heptane were carried out separately in a low pressure chamber Langfang (altitude 50 m) and in airport laboratories Kangding (altitude 4290 m) at the air pressure of 60 kPa according to ISO-9705. The main characteristic parameters were burning rate, flame temperature, and radiation heat flow. Increased oxygen concentration caused the burning rate of the material to increase and burning time to decrease. A higher amount of air absorbed, or higher oxygen concentration, could increase the fire intensity in the cargo area and worsen the flame spread rate.”

is removed from 1.2., and it can be included in Chapter 1, after:

„Upholstered seats represent a part of the interior equipment of an aircraft. There are separate tests relating to these materials [8,9].” (my earlier suggestion).

2) 208, 209:  Each picture was made at actual time,  which is written under it (Fig. 3).

Author Response

We thank the opponent for the helpful comments. We corrected comment 1) and 2) in the paper.

Reviewer 2 Report

The manuscript is improved but still suffers from awkward sentence structure and an overuse of author's names and personal pronouns.

Author Response

We thank the opponent for the helpful comments.

This manuscript is a resubmission of an earlier submission. The following is a list of the peer review reports and author responses from that submission.

Round 1

Reviewer 1 Report

This work describes the assessment of three different materials used in commercial seats under fire conditions. I would recommend the paper to get rejected for the following two reasons.

  • The text contains several grammatical and syntax errors, already from the abstract. It seems that the authors have either not put a considerable effort to revise the language, or a native speaker needs to make revision. Some examples in the abstract only: Line 15: frequent should be frequently. Line 18: focused on and not focused of. Line 19 misses a verb. Also, already the first line of the introduction does not form a meaning. This continues throughout the text.
  • Overall the whole study is looking more like a technical report which is used to evaluate some specific materials intended for commercial use. This is also evident from the conclusions. I do not find significant scientific impact in this paper.

Finally, I do not find a lot of meaning in Figure 1. We just practically see three different colors. Maybe a higher resolution image to show some insight in the texture or some more information on the chemical composition would be more important. However, part of chemical composition is also given in Table 2.

Author Response

We thank the opponent for the helpful comments. We have tried to carefully and consistently modify the article to improve its quality and offer our experimental results. We are in research team, which deal with research of furniture materials. Our tasks, which are cigarette test and simulated equivalent of flame indicates flammability but cannot provide reliable information about the flammability course of the final product (furniture).

We accepted your comments and we make the new revision of the English by native speaker.

We included figure 1 with higher resolution.

Reviewer 2 Report

Comments on the manuscript "Fire Characteristics of Textile Materials Used in  Upholstered Seats":

This paper presents the fire tests on some frequently used materials for upholstery. The results were analyzed by ANOVA.

In the opinion of the reviewer, this manuscript lacks scientific contributions.  Both the tests and analysis are very simple. The reviewer suggests being declined.

Author Response

We thank the reviewer for the comments. We have tried to carefully and consistently modify the article to improve its quality and offer our experimental results with statistic evaluation. We are in research team, which deal with research of furniture materials. Our tasks, which are cigarette test and simulated equivalent of flame indicates flammability but cannot provide reliable information about the flammability course of the final product (furniture). We have improved the scientific value and also make more details test by statistical program.

Reviewer 3 Report

This manuscript reports the results of an evaluation of the ignitability of materials used in construction of upholstered seats, particularly for transportation. Useful results are reported. However, discussion is quite limited.

The manuscript will need some revision for clarity and readability. Corrections are penciled-in directly on pages of the manuscript attached. These are representative of the kinds of changes needed throughout. Personal pronouns, author's names and et.al. should be omitted.

Author Response

We thank the reviewer for the helpful comments and correction. We have tried to carefully and consistently modify the article to improve its quality.

Reviewer 4 Report

  1. The subject of the paper is very important. The paper is a preliminary recognition of fire characteristics of upholstery materials in seats. Results of the research presented in the paper indicate the need to continue the research in a broader aspect (upholstery and interior of seats). In future, the subject matter requires widening of knowledge of materials in seats, after contact with the source, which can lead to the burning of the material. It is important to accurately specify of researches conditionsconduct and input parameters (e.g., material composition, ambient oxygen concentration) and output parameters (e.g., amount and rate of heat release, amount and type of compounds released, flame propagation rate...) .
  2. I suggest changing the topic: Fire characteristics of upholstery materials in seats
  3. Please complete Keywords (“…The aim of this article is analysis and comparison of the effect of initiation sources (cigarettes and matches) on the upholstery materials…”)
  4. Authors write: 1.1. “…Upholstery is the surface treatment of materials and walls using textile, leather, leatherette for, netting or other natural and synthetic materials [7]. Upholstery is composed of external covering (outer cover), of inner liner (inner cover) (coating thicker than 2mm), and padding (filling). Cushioning belongs to the category of materials which are easily flammable, such as latex foam, polyurethane foam, etc. These materials react dangerously in contact with fire: they melt, run and release toxic residues…”.....„…The test determines only the flammability of material combinations which the upholstered sitting furniture consists of, and does not determine the flammability of individual materials used for its manufacture. The test indicates its flammability but cannot provide reliable information about the flammability course of the final product (furniture) [7]…”          This information is very important and it should be included (in brief) in Abstract and Conclusions of paper, with indication, that results of the research presented in this paper indicate the need to continue the research in a broader aspect.
  5. 1. Textile materials in seats and 1.2. Textile materials in seats are the same. I suggest changing: 1.2. Sources of combustion initiation
  6. Authors write: 2.2. “…The burning course is monitored and any traces of sustained smoldering or flame combustion of the padding or cover are recorded…” In Chapter 2.2., 2.3 please describe (in more detail) the measurement methodology, taking into account the registration of test results, and include the results of this registration (graphs?) - in Chapter 3.1., 3.2.
  7. Text: in 3.1. “The energy released from the cigarettes proves itself as not very effective………….In 2012, Lloyd treated this issue [36] while presenting the diversity of cigarette position and the time of application on wheelchair materials.”, in 3.2. “Wang et al. [37] simulated the conditions of fire in the cargo area of an aircraft. ………. A higher amount of air absorbed, or higher oxygen concentration, could increase the fire intensity in the cargo area and worsen the flame spread rate.”  is a literature review and should be in the Chapter 1.
  8. Pictures in Fig. 3 (a, b, c) should show the same test (for example Fig. 3a: picture 1,2,3;  3b: picture 1,2,3) at appropriate intervals of time of this process (please add time of process for each drawing: 3a-1,2,3, 3b-1,2,3, 3c-1,2,3). Please wider discuss the obtained research results in Chapter 3.1. and 3.2.
  9. Please insert the list of symbols and signs.
  10. Please wider describe the methodology in Chapter 3.3., and determine values in Table 6.
  11. Discussion - should be Conclusions
  12. Please standardize References (according to the magazine's guidelines)
  13. [19] - why red

Author Response

We thank the reviewer for the helpful comments and correction. We have tried to carefully and consistently modify the article to improve its quality and offer our experimental results with corrected statistical evaluation. Our changes are highlight directly in manuscript: yellow – is correction of text and tables, green – new added text, blue is correction of references.  We have the following comments for each request:

Point 1: Thank you for the helpful comments we have incorporated into the article. We believe that editing the article has made the article better.

Point 2: We agree with title and we made change

Point 3: We added key words

Point 4 We agree with reviewer ´s comments and we added in Abstract and in Conclusion the text about: “However, it is generally know, that cigarette test indicates flammability but cannot provide reliable information about the flammability course of the final product (furniture).“

Point 5: We modified this subtitles

Point 6: Yes, we monitored, But, our monitoring of experiments were visual with chronological changes by the time. We had 1 thermocouple under tested sample and we wanted know temperature in flesh ignition but this part experiments this set of experiments was imperfect and the results biased.

Point 7: We changed position Llyod and Wang et al. Chapter 1.

Point 8: Thanks form motivated comments. We revised Chapter 3 Results

Point 9: We added List of symbols and signs.

Point 10: We described the methodology in Chapter 3.3., and determine values in Table 6.

Point 11: We changed Conclusions for Discussion

Point 12: We standardized References

Point 13: Sorry. It was mistake.

Round 2

Reviewer 1 Report

I still find that the language needs improvement. Also, my overall impression that the findings are not so significant still remains. I had recommended that the paper gets rejected and still after the corrections I do not see a valid justification for further consideration regarding publication.

Reviewer 2 Report

The reviewer still suggests rejecting. The scientific contributions of this manuscript are not suitable for this journal.

Reviewer 4 Report

1) I propose:

Keywords: initiation sources of ignition (cigarettes, matches); upholstery materials in seats

2) I propose change to:

Abstract:

„The article deals with selected upholstery flammability test materials that, in the case of fire, can cause fire spread. For the research, the frequently used materials for upholstery based on of polyester was utilized: imitation leather, suede and microplush. Initiation of initiating spontaneous flammability with various sources of ignition were measured a smoldering cigarette and a match flame. Results were measured in treatment of smouldering time and length of the burnt-though sample. Upholstery materials an integral part of seat construction. To be used in transport, upholstered material must meet safety measures such as the strength, sanitariness, and fire resistance. All tests were performed in accordance with applicable technical standards. Impact assessment of the sample (weight) on "smoldering time" and "length of degradation" was carried out using ANOVA statistical analysis. Significant differences in "length of degradation" can be observed between samples. Tests cannot provide reliable information about the flammability course of the final product. Upholstery is composed of external covering, of inner liner, and padding. Results of the research presented in this paper indicate the need to continue the research in a broader aspect.”

3) I propose change in Chapter 1:

"...Upholstered seats represent a part of the interior equipment of an aircraft. There are separate tests relating to these materials [8,9].

The specific sources of ignition for the manufacture of upholstered sitting furniture are smoldering cigarette and equivalent of flame by matches. This sources are included in technical laws for flammable testing of upholstered sitting furniture [8,9]. We need to note that objections to the existing methods of the tests of upholstery material have been made. In 2012, Lloyd treated this issue [23] while presenting the diversity of cigarette position and the time of application on wheelchair materials. These tests are part of a wealth of research on passenger safety. Tests do not provide the end result of product safety, but as partial research they are the foundation of building a safe product for the passenger. Wang et al. [24] simulated the conditions of fire in the cargo area of an aircraft. He studied behavior characteristics of the selected materials in a partially closed chamber and in an alpine laboratory for the same low atmospheric pressure. He observed the control effects of exhaust air and oxygen concentration. Experiments with fire using n-heptane were carried out separately in a low pressure chamber Langfang (altitude 50 m) and in airport laboratories Kangding (altitude 4290 m) at the air pressure of 60 kPa according to ISO-9705. The main characteristic parameters were burning rate, flame temperature, and radiation heat flow. Increased oxygen concentration caused the burning rate of the material to increase and burning time to decrease. A higher amount of air absorbed, or higher oxygen concentration, could increase the fire intensity in the cargo area and worsen the flame spread rate." - text please remove from 1.2.

4) It seems to me, that:

- Figures IIa and IIIa are not a continuation of the process in Fig. Ia

- Fig. IIIb is not a continuation of the process in Fig. IIb

5) The text:

"The energy released from the cigarettes proves itself as not very effective, with no flame initiation in most cases [35]. The cigarette burning is, in the process of smoldering, accompanied by the release of harmful products. Their characteristics are presented in the works of [36].

Stoliarov [37] explains the processes as follows: „The pyrolysis region, dominated by anaerobic decomposition, provided gaseous fuel, the ignition of which resulted in the transition. The smoldering region, dominated by oxidation reactions at the solid-gas interface, generate the heat necessary to maintain the pyrolysis process and ignite the gaseous fuel“ [37].

We need to note that objections to the existing methods of the tests of upholstery material have been made. In 2012, Lloyd treated this issue [23] while presenting the diversity of cigarette position and the time of application on wheelchair materials. "

is a literature review and should be too in the Chapter 1.

6) Text:

"In 2012, Lloyd treated this issue [23] while presenting the diversity of cigarette position and the time of application on wheelchair materials."

is repeated in the article.

7) In Chapter 3.3. please insert an additional Table with values of results of experiments, that have been included in the analysis (in Tables 5 and 6).

Please include the number of experiments replicates.

8)  I propose add the text - at the end of Conclusions

"The research presented in this article needs to be continued in the future."